# Knowledge of behavioral risk factors for type 2 diabetes mellitus and its associated factors among women of reproductive age

Tinsae Seyoum[1]☺*, Selamnesh Tesfaye[1]☺, Yohannes Shiferaw[1‡], Rahel Hailu[1‡], Dagim Tefera[1‡], Zeleke Gebru[2‡]

**1** Department of Public Health, Arba Minch College of Health Sciences, Arba Minch, Ethiopia, **2** Departments of Public Health, Arba Minch University College of Medicine and Health Sciences, Arba Minch, Ethiopia

☺ These authors contributed equally to this work.
‡ YS, RH, DT and ZG also contributed equally to this work.
* tswy2006@gmail.com

**Data Availability Statement:** All relevant data is within the manuscript and its supporting information files.

## Abstract

### Background

Type 2 diabetes accounts for over 90% of all diabetes cases and is caused by a combination of behavioral risk factors. It is currently a serious health issue, particularly among women of reproductive age, as it is associated with reproductive disorders. Preventing it requires knowledge, but there is limited data on behavioral risk factors in Ethiopia.

### Objective

To assess knowledge of the behavioral risks of type 2 diabetes mellitus and its associated factors among women of reproductive age.

### Methods

A community-based cross-sectional study was conducted, with all women in the town serving as the source population. A multistage sampling method was utilized to recruit kebeles, and a systematic random technique was employed to select households at every 13th interval. We completed interview questionnaires for 623 samples. The crude odds ratio was calculated using a bivariate logistic model, and multivariate analysis was performed to control for confounding and identify associated factors among model-fitting variables using an adjusted odds ratio (AOR).

### Result

The knowledge of behavioral risk factors (BRF) among women of reproductive age (WRA) is 47.0% [95% CI, 43.5–50.9], and significant associations were found with the following factors: average family income of between 3000 and 5000 Ethiopian Birr(ETH) 1.81 [95% CI, 1.03–3.18], > = 5001 ETH 1.93 [95% CI, 1.02–3.68], diabetes mellitus (DM) in the friend or relatives 4.03 [95% CI, 1.56–10.46], family history of DM 9.47 [95% CI, 4.74–18.90], source

**Funding:** NO-Arba Minch Health Science College has given us a fund to do our research but has no role in study design, data collection and analysis, decision to publish, or even preparing the manuscript.

**Competing interests:** The authors have declared that no competing interests exist.

**Abbreviations:** AMCHS, Arba Minch College of Health Science; BRF, Behavioral Risk Factors; CI, Confidence Interval; DM, Diabetes Mellitus; ETH, Ethiopian Birr; T2-DM, Type 2 Diabetes mellitus; WRA, Women of Reproductive Age.

of information: health workers 1.87 [95% CI, 1.04–3.34] and friend or relatives 1.65 [95% CI, 1.04–2.62].

## Conclusion

Knowledge of behavioral risk factors for type 2 diabetes was poor among study participants. Factors such as family income, diabetes mellitus (DM) in friends or relatives, family history of DM, and sources of information were strongly associated with good knowledge. It is essential to emphasize health education about behavioral risk factors for women.

## Introduction

Diabetes mellitus (DM) is one of the main types of non-communicable diseases caused by a combination of different factors [1, 2]. Type 2 diabetes (T2-DM) is a non-insulin-dependent condition characterized by inadequate production and peripheral resistance to insulin [3]. It accounts for over 90% of all types of diabetes in the world and usually occurs in middle-aged and older people; however, it progressively affects the most productive age groups [4]. People with diabetes have a high risk of developing serious health problems that can lead to a reduced quality of life [5]. High blood glucose levels could also disrupt different vital parts of the body, including the heart, kidneys, eyes, and nerves, resulting in various vascular complications [6]. In addition, the progression of high blood glucose levels during reproductive age may heighten the risks for future generations [7]. It is also recognized as a contributing factor to the challenge of sustainable development [1].

T2-DM is experiencing rapid global expansion due to several factors, including aging populations, economic development, and urbanization. These changes are all associated with BRFs. The incidence of diabetes also increases gradually during puberty, and it is more prevalent in women, possibly due to hormonal variations and insulin resistance [4]. Additionally, it is linked to reproductive issues in individuals of reproductive age, which can be addressed through behavioral changes [7]. Around 451 million people over the age of 18 have diabetes, and nearly 22 million women's live births are affected by the prenatal period [8].

It also affects fourteen percent of all live births in the Middle East and Africa [9]. Preventing diabetes at the population level relies heavily on public awareness and a proper understanding of its risk factors [10].

Multiple factors contribute to the development of T2-DM, categorized as non-modifiable and modifiable risk factors. The modifiable risk factors known as BRFs that can increase the likelihood of developing T2-DM include obesity, physical inactivity, unhealthy eating habits, smoking, excessive alcohol consumption, insomnia, and stress [11, 12]. These issues often remain hidden in countries with low socioeconomic status, including ours [13, 14].

BRF significantly predisposes individuals to T2-DM. However, adopting positive lifestyle changes and reducing negative behaviors can reflect an individual's knowledge of these risks [15]. Although different studies have assessed knowledge of certain risk factors like sedentary lifestyle, obesity, poor diet, and lack of exercise in diabetes and chronic disease patients, they have overlooked other factors. This study considers all relevant factors and addresses the limited understanding of knowledge regarding BRF [10, 15–22]. In Ethiopia, research has predominantly concentrated on the general understanding of DM, with limited evidence on the knowledge regarding BRFs [23–25].

The benefit of this study is improving health outcomes by identifying women's awareness and increasing their knowledge of BRF. It aims to target interventions and inform policy development. The beneficiaries include WRA, health care providers, policymakers, families, and communities.

Therefore, this study assesses WRA's understanding of all potential BRFs for T2-DM and associated factors. This study can also serve as a base for interventions, improve knowledge on healthy lifestyle modifications and T2-DM prevention, and be a reference for scholars.

## Materials and methods

A community-based, cross-sectional quantitative study was conducted in Arba Minch town from February 20, 2022, to March 22, 2022.

According to the 2014 national population and housing census, Ethiopia has a female population of 46.9 million, which accounts for 49.8% of the total population. In the study area, the estimated number of WRA is 31,560, which represents 23.3% of the total population.

The source population consisted of all women living in Arba Minch town. The study population included WRA between 15 and 49 who had resided in the selected kebeles for at least six months. Women who were severely ill and mentally ill and also unresponsive during data collection were excluded from the study.

To conduct the study, a multi-stage sampling method was used. Four kebeles out of the eleven in the town were randomly selected using a random sampling technique. The required sample size was then proportionally allocated based on the number of households in each selected kebele.

Before data collection, a census was undertaken to identify the eligible population in the cho-sen kebele.

The sample size determination was calculated using a single population formula: $n = \frac{Z(\alpha/2)^2 P(1-P)}{d^2}$ Where n = the desired sample size, Z = standard deviation at the 95% confidence level (1.96).

P is the proportion in the target population estimated to have a particular characteristic. therefore, the proportion of knowledge of risk factors for DM in Nigeria was 75.5% [26].

d = the allowable error of margin is 5% = 0.05.

Therefore, $n = \frac{((1.96)^2 * 0.755 * (1-0.755))}{(0.05)^2} = 284.23.$

Using this formula, the sample size was 284, then multiplied by the design effect of 2 and added 10% of the non-response rate, resulting in a final sample size of 625.

The sample size was determined using a multistage sampling method, which allocated participants proportionally based on the population size of the selected kebeles. After obtaining the number of households from the census, every 13th household was selected until the required sample size was achieved. The first house was selected using a lottery method. If multiple members of a household met the inclusion criteria, only one individual was chosen through the lottery.

In this study, knowledge of BRF is the dependent variable. The independent variables include general knowledge of T2-DM (such as its meaning, symptoms, prevention, and control), behavioral risk perception, leading a healthy lifestyle, socio-demographic factors, and other relevant factors encompass personal history of diabetes, family history of diabetes, exposure to diabetes health education, awareness of diabetes, and sources of information.

## Operational definitions

**Behavioural risk factors.** Factors that increase the likelihood of T2-DM and can be modified or controlled. These include physical inactivity, an unhealthy diet, harmful alcohol consumption, smoking, obesity, and stress [8, 16, 17].

**Knowledge of behavioral risks.** Individuals answering behavioural risk-related questions below the mean value will be considered to have poor knowledge, while those who answer above the mean will be deemed to have good knowledge [15, 22, 27].

## Data collection

All questions were translated from English to Amharic and then translated back to ensure accuracy. The study instrument included questions on demographics and knowledge of BRF (physical activity, obesity, diet, tobacco use, stress, and sleep time). Standardized training for the attending data collectors was also given. Data collectors were degree holders and health professionals. A pretest was conducted with 5% of the sample population who lived outside the study area (Mirab Abaya town). The questionnaire's validity and consistency were tested, and small changes were made based on the pretest.

The questionnaire's internal consistency was assessed by measuring the Cronbach alpha coefficient for questionnaire scales, and the test results were 0.79 (*r = 0.787*).

After obtaining consent from respondents over the age of 18, as well as from the parents and guardians of the minors included in the study, face-to-face interviews were conducted with the selected women. This approach allowed time for any questions to be clarified as needed. Each interview lasted an average of 25 minutes.

## Data quality management

For data quality measures, training was given to all data collectors and supervisors on their duties and the purposes of the study. A pretest was conducted outside the study area in a similar environment to prevent information contamination. Modifications were made to the questionnaire, and all supervisors and investigators adhered to reports to ensure data completeness and provide further clarification. Double data entry was done in the Epi-data to prevent data entry errors, and the data was properly labeled and coded during the data cleaning phases.

## Data processing and analysis

Data entry was performed using Epi Data version 4.6, while all statistical analyses were conducted with SPSS version 21. At the end of each interview and before the study was completed, data editing was carried out to ensure the completeness of the questionnaire. Each correct response was assigned a score of 1, whereas incorrect or uncertain responses received a score of 0. The questions related to BRF were scored individually, and the scores were summed to achieve a maximum total score of twelve [12]. A mean score of less than 4.60 was classified as poor knowledge, while a score above 4.60 indicated good knowledge of BRF for T2-DM.

Descriptive statistics was used to summarize and describe the features of the data, and binary logistic regression analysis was applied to assess the association between variables. The variables with p-values less than 0.25 were transferred into multivariate analysis to control the possible confounders. Those variables with a p-value of < 0.05 in the multi-variable analysis were considered to have a significant association with the outcome variable. The Hosmer and Lemeshow test was used to check the assumption tests and models of goodness of fit (P-value = 0.160).

A test for multicollinearity among the independent variables was performed using the variance inflation factor (VIF). A cutoff point of 5 was established, with a tolerance level greater

than 0.1. The correlations observed were within acceptable limits. The study results are presented in tables, figures, and accompanying text.

### Ethical considerations

This study received approval from the review board of Arba Minch College of Health Sciences. An ethical clearance letter (Ref No/አ/ም/ጤ/ሳ/ክ/01/20/3587) was obtained from the Ethical Review Committee of the AMCHS. A letter from the Research Ethics Committee was submitted to the respective offices. During data collection, informed verbal and written consent was obtained from respondents (those above 18 years old), and assent from family members and guardians was obtained for those respondents whose age is between 15 and 17 years after explaining the purpose of the study. To minimize the transmission of COVID-19 as much as possible, the data collectors strictly followed the procedure.

## Results

### Socio-demographic and socio economic characteristics of the respondents

Of the total of 625 calculated study participants, 623 agreed to participate in the study; with this, the response rate is 99.68%. The minimum age of respondents is 15 years old, and the maximum age is 49 years old. The mean age of respondents was 27.3 (SD ± 8.2). Of the respondents, 359 (57.6%) were married. The study found that 27 (4.3%) individuals were illiterate and able to write and read, while 235 (37.7%) of participants had education at the college level and above. One hundred eighty-one (29%) were students. The average family income per month was 3910.00 ETH (SD ± 3107.91) (see Table 1).

The study revealed that 568 (91.2%) participants were aware of diabetes, with 264 (42.2%) having information from their friends or relatives, 159 (25.4%) receiving health education about T2-DM, and 192 (31%) having their friends or relatives with diabetes; besides this, 134 (21.5%) had diabetes in their families (Table 2).

### General knowledge of type-2 DM

**Knowledge of T2-DM meaning.**   In terms of diabetes knowledge, Table 3 reveals that 511 participants (82.0%) were unaware that T2-DM could result from insufficient insulin use. Additionally, 535 participants (86.0%) did not recognize that T2-DM could arise from an improper response to insulin. Among the participants, 248 (39.8%) understood that diabetes is characterized by high blood sugar levels. Approximately half of the participants, 312 (50.1%), were aware that diabetes is incurable, while 333 (53.6%) were uncertain whether the condition affects all parts of the body. Overall, the total rate of correct responses was 29%.

**Knowledge of risk factors for T2-DM.**   Among the study participants, 345 (55.4%) mentioned old age, and 307 (52.8%) also mentioned heredity as a non-modifiable risk factor for T2-DM.

In the study, 354 (56.8%) and 351 (56.3%) participants noted excessive sugary, sweet foods and poor dietary habits as risks for T2-DM. Nearly half of the participants, 409 (49.3%), also stated that obesity could be a risk factor, and 275 (44.1%) reported that all T2-DM risk factors could be preventable. Among participants, 264 (42.4%) mentioned being hypertensive, and 251 (41.3%) did not get enough activity and exercise as risk factors for T2-DM. Additionally, the lowest number of respondents reported a sedentary lifestyle 203 (32.6%), insomnia 135 (21.7%), and gestational diabetes 121 (19.4%) as a risk factor (see Table 4).

**Table 1. Socio-demographic status of respondents on behavioral risk factors of type-2-diabetes mellitus in Arba Minch town 2022 [N = 623].**

| Variables | | Frequency | Percentage |
|---|---|---|---|
| **Age** | 15–24 year | 243 | 39 |
| | 25–30 year | 177 | 28.4 |
| | 31–40 year | 126 | 25.4 |
| | > = 41 year | 77 | 7.2 |
| **Marital status** | Single | 232 | 37.2 |
| | Married | 359 | 57.6 |
| | Separate/divorce | 16 | 2.6 |
| | Divorce | 16 | 2.6 |
| **Educational level** | Illiterate | 27 | 4.3 |
| | Read and write | 27 | 4.3 |
| | Grade 1–4 | 35 | 5.6 |
| | Grade 5–8 | 121 | 19.4 |
| | Grade 9–12 | 178 | 28.6 |
| | Collage and above | 235 | 37.7 |
| **Occupation** | House wife | 164 | 26.3 |
| | Student | 183 | 29.1 |
| | Governmental employee | 128 | 20.5 |
| | Merchant | 84 | 13.5 |
| | Private employee | 26 | 4.2 |
| | Daily labor | 19 | 3 |
| | unemployed | 19 | 3 |

The level of knowledge of BRF for T2-DM among reproductive age women was computed, and the level of good knowledge was 47% (mean score $\geq$ 4.60 with SD 3.46) (see Fig 1. Level of knowledge of BRF).

**Table 2. The description of response of study participant on the socioeconomic and awareness status in Arba Minch, 2022 (n = 623).**

| Variables | Response | No | percent |
|---|---|---|---|
| **Average family income** | <1999 ETH | 161 | 25.8 |
| | 2000-2999ETH | 91 | 14.6 |
| | 3000–5000 ETH | 232 | 13.5 |
| | >50001 ETH | 139 | 22.3 |
| **Heard about DM** | Yes | 568 | 91.2 |
| | No | 55 | 8.8 |
| **Source of information N = 568** | Media | 185 | 32.5 |
| | Health workers | 116 | 20.5 |
| | Friend /relatives | 267 | 47 |
| **DM health education** | Yes | 159 | 25.5 |
| | No | 464 | 74.5 |
| **Diabetes in friend /relatives** | Yes | 192 | 30.8 |
| | No | 395 | 63.4 |
| | I don't know | 36 | 5.8 |
| **Family with DM** | Yes | 134 | 21.5 |
| | No | 458 | 73.5 |
| | I don't know | 31 | 5 |

**Table 3. The description of general knowledge about type-2-DM definition among women of reproductive age in Arba Minch, 2022.**

| Variable | Yes | | No | | I don't know | |
|---|---|---|---|---|---|---|
| | No | % | No | % | No | % |
| Define T2-DM | | | | | | |
| Insufficient use of insulin | 57 | 9.10 | 55 | 8.80 | 511 | 82.00 |
| Not responding for insulin | 58 | 9.30 | 30 | 4.80 | 535 | 85.90 |
| High level of sugar | 248 | 39.8 | 41 | 6.60 | 334 | 53.60 |
| Not curable | 312 | 50.10 | 162 | 26.00 | 149 | 23.90 |
| Affect any part of the body | 230 | 36.90 | 59 | 9.50 | 334 | 53.60 |

**Knowledge of symptoms of type 2-DM.** Regarding the symptoms of diabetes, 416 (66.8%) respondents mentioned excessive hunger, 324 (52%), a high level of blood sugar, and 311 (50%) excessive thirst. The least well-known symptom among respondents was blurred vision, accounting for 199 (31.9%) of the total knowledge of symptoms. (see Fig 2. Knowledge of symptoms).

**Knowledge relate to prevention of DM and its complication.** Table 5 displays responses to questions concerning diabetes prevention. Of all respondents, 458 (73.5%) mentioned preventing diabetes complications with available drugs. More than half of the participants, 318 (51%), also knew about consuming a healthy diet (more fruits and vegetables) every day, and 47 percent said proper weight reduction and regular exercise could help prevent DM and its complications. The respondents' average knowledge of T2-DM prevention and its complications was 55%.

## Behavioral risk perception for type 2DM

The study found that, out of 623 participants, 428 (68.7%) agreed to undergo diabetes testing, and 387 (61.1%) felt that their family members should be checked for diabetes. Around 387

**Table 4. The frequency distribution of participants response on knowledge of behavioral risk factors for type 2 diabetes mellitus, Arba Minch town, 2022 (n = 623).**

| Variable | Yes | | No | | I don't know | |
|---|---|---|---|---|---|---|
| | No | % | No | % | No | % |
| What are the risk factors of T2-DM? | | | | | | |
| Old age * | 345 | 55.4 | 108 | 17.3 | 170 | 27.3 |
| Hereditary * | 329 | 52.8 | 94 | 15.1 | 200 | 32.1 |
| Obesity ** | 307 | 49.3 | 77 | 12.4 | 239 | 38.4 |
| Gestational diabetes (Pregnancy) ** | 121 | 19.4 | 129 | 20.7 | 373 | 59.9 |
| Sedentary life ** | 203 | 32.6 | 70 | 11.2 | 350 | 56.2 |
| Poor dietary habits (not including fruits and vegetables) ** | 351 | 56.3 | 63 | 10.1 | 209 | 33.5 |
| Not getting enough activity ** | 251 | 41.3 | 84 | 13.5 | 282 | 45.3 |
| Stress ** | 227 | 36.4 | 87 | 14.0 | 309 | 49.6 |
| Hypertension ** | 264 | 42.4 | 81 | 13.0 | 278 | 44.6 |
| Sleeplessness ** | 135 | 21.7 | 108 | 17.3 | 380 | 61.0 |
| Smoking cigarette ** | 149 | 23.9 | 101 | 16.2 | 373 | 59.9 |
| Heavy Alcohol consumption ** | 223 | 35.8 | 84 | 13.5 | 316 | 50.7 |
| Eating too much sugar and sweet ** | 354 | 56.8 | 75 | '12.0 | 194 | 31.1 |
| All risks are preventable ** | 275 | 44.1 | 23 | 3.7 | 325 | 52.2 |

*Non modifiable risk factors

** Modifiable risk factors or behavioral risk (BRF)

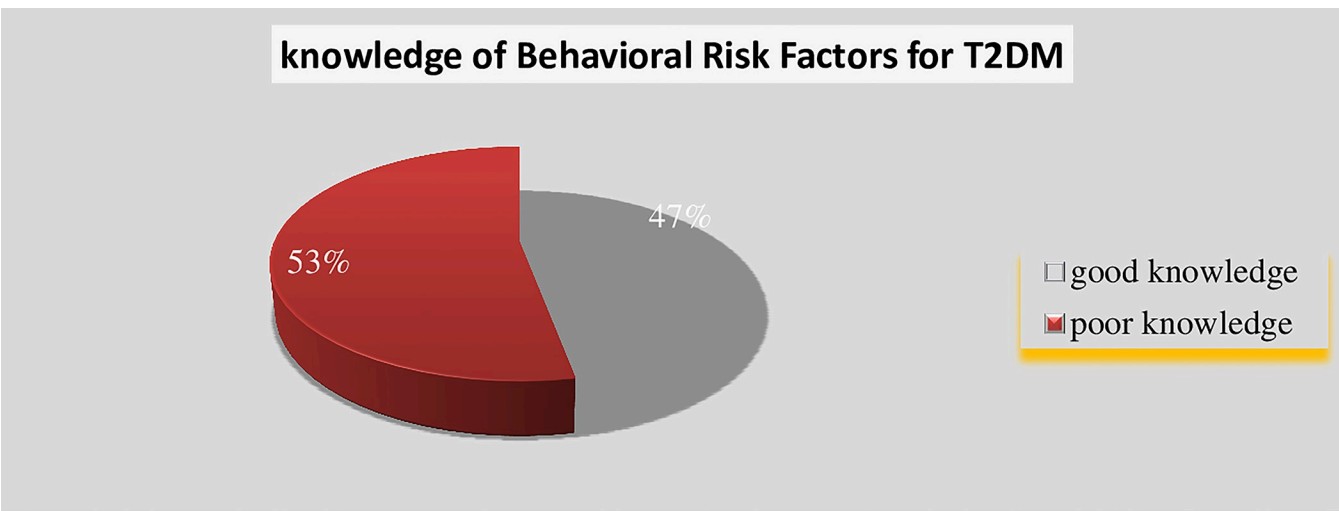

**Fig 1. The level of knowledge of behavioural risk factors for type-2DM among reproductive age women in Arba Minch town, 2022.**

(62.1%) believed that avoiding fatty and sugary foods helped control diabetes, whereas 274 (53.3%) believed that physical activity helped prevent it. Overall, 368 (59%) of respondents had perceived the risk assessment (see Table 6).

### Adopting a healthy life style for type 2 DM

Regarding adopting a healthy lifestyle, 235 (37.7%) of participants were involved in healthy lifestyles that promote diabetes impediment and are crucial for diabetes prevention. Only 49 respondents (7.9%) regularly consume high levels of fatty foods (saturated fats) and sugary foods, which can contribute to diabetes mellitus (DM). Additionally, 49 respondents (7.9%) engage in regular exercise for weight maintenance. However, only 17 individuals (2.7%)

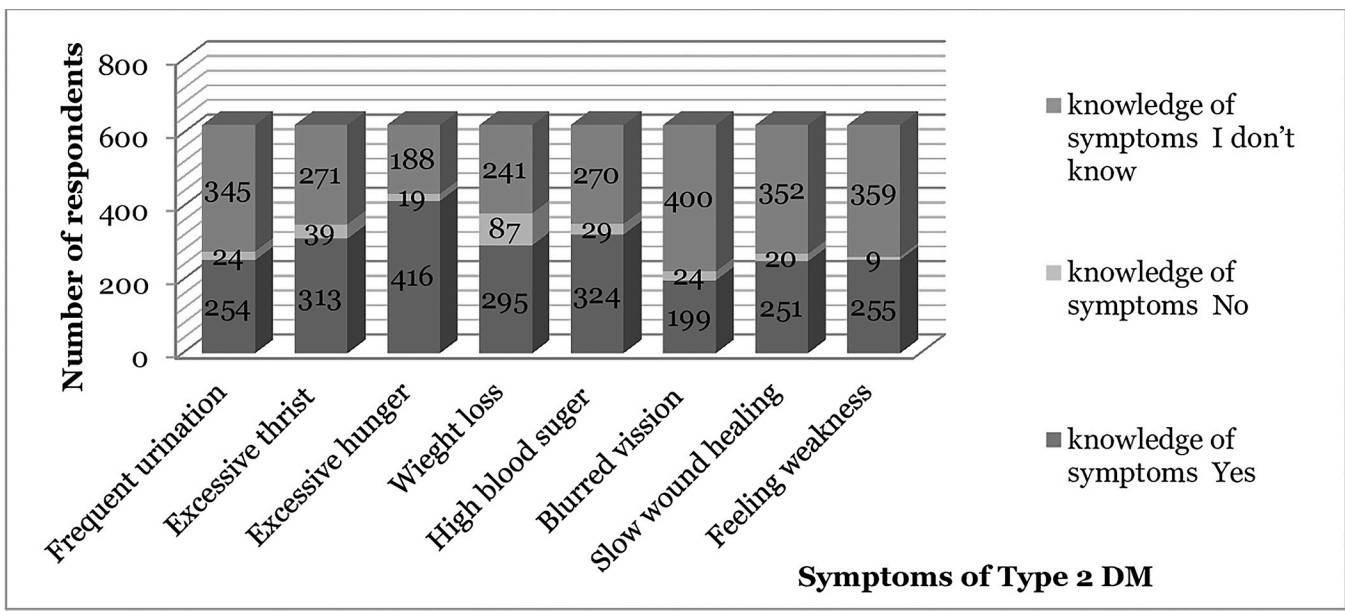

**Fig 2. The response of reproductive age women on their knowledge of symptoms for T2DM in Arba Minch, 2022 (N = 623).**

**Table 5. The knowledge of complication prevention of type 2-DM among reproductive age women in Arba Minch town, 2022 (n = 623).**

| Variable | Yes | | No | | I don't know | |
|---|---|---|---|---|---|---|
| | No | % | No | % | No | % |
| **Drugs are available to prevent complication** | 458 | 73.50 | 24 | 3.90 | 141 | 22.60 |
| **Regular Exercise can help prevent complication** | 298 | 47.80 | 95 | 15.20 | 230 | 56.20 |
| **Consuming healthy diet help prevent complication** | 318 | 51.00 | 99 | 15.60 | 206 | 33.10 |
| **Proper weight reduction prevent complication** | 291 | 46.70 | 107 | 17.20 | 225 | 36.10 |

consistently monitor their blood sugar levels, and 27 respondents (4.3%) regularly check their blood pressure. Among the participants, 477 people (76.6%) do not consume any alcohol, and almost all participants, 621 individuals (99.7%), do not smoke cigarettes (see Table 7).

## Factors associated with knowledge of behavioral risk factors for type 2-DM

A bi-variate logistic regression analysis indicated that several factors, including age, educational status, occupation, monthly family income, awareness of diabetes mellitus (DM), information source, family history, and having friends or relatives with DM, were statistically associated with knowledge about diabetes risk factors (BRF) (Table 8). Consequently, variables with a P-value of less than 0.25 were included in a multivariate logistic regression model. Ultimately, age, educational status, occupation, and hearing about of DM were found to be non-significant.

The average monthly family income ranges from 3,000 to 5,000 ETH. are 1.81 times more likely (AOR = 1.81, 95% CI = 1.03–3.18) to have good knowledge about BRF compared to those with a monthly income of less than 1,999 ETH. Households with an average monthly income exceeding 5,001 ETH are also 1.93 times more likely (AOR = 1.93, 95% CI = 1.02–3.68) to possess good knowledge about BRF than those in lower income.

Individuals who have a friend or relative with a history of DM are 4.03 times more likely (AOR = 4.03, 95% CI = 1.56–10.57) to be knowledgeable about BRF compared to individuals who are unaware of any such histories. Furthermore, those with a family history of DM are 9.47 times more likely (AOR = 9.47, 95% CI = 4.74–18.90) to show an association with knowledge about BRF than those without a family history of the disease. Additionally, individuals who received information about BRF from friends or relatives are 1.65 times more likely (AOR = 1.65, 95% CI = 1.04–2.62) to have knowledge about it than those who obtained their information from media sources. Women who received information about BRF from health workers are 1.87 times more likely (AOR = 1.87, 95% CI = 1.04–3.34) to be knowledgeable about it than those who received information from the media.

A weak correlation was observed among each independent variable. The independent variables do not have a strong linear relationship with each other. In other words, changes in one variable are not strongly associated with changes in another variable.

**Table 6. The perception of reproductive age women on behavioral risks of type 2-DM in Arba Minch town, 2022 (n = 623).**

| Variable | Agreed | | I Don't know | | Disagreed | |
|---|---|---|---|---|---|---|
| | No | % | No | % | No | % |
| **You should be examined for DM?** | 428 | 68.70 | 142 | 22.80 | 53 | 8.50 |
| **Family members should be screened?** | 387 | 61.10 | 196 | 31.50 | 40 | 6.40 |
| **You should follow avoiding too much sugar?** | 332 | 57.80 | 256 | 36.80 | 35 | 5.50 |
| **Physical activity can prevent the risk?** | 274 | 53.30 | 328 | 41.10 | 21 | 5.60 |
| **Maintaining a healthy weight is important?** | 268 | 43 | 330 | 53 | 25 | 4.0 |
| **Early medical follow up can save life earlier?** | 445 | 71.4 | 158 | 25.4 | 20 | 3.2 |

**Table 7. The description of behavioral risk and adoption of health status of women reproductive age for type 2-DM in Arba Minch town 2022 (n = 623).**

| Variable | Frequently | | Occasionally | | Never | |
|---|---|---|---|---|---|---|
| | No | % | No | % | No | % |
| Do you consume of fatty and sweets food? | 49 | 7.9 | 472 | 75.8 | 102 | 16.4 |
| Do you do 30–60 mints physical activity daily? E.g. Brisk walking, house activities, climbing staircase. | 188 | 30.2 | 314 | 50.4 | 121 | 19.4 |
| Do you participate in maintaining your healthy weight? | 49 | 7.9 | 379 | 61 | 195 | 31.3 |
| Do you check your blood sugar regularly? | 17 | 2.7 | 177 | 28.4 | 429 | 68.9 |
| Do you check your blood pressure regularly? | 27 | 4.3 | 225 | 36.1 | 371 | 59.6 |
| Do you drink alcohol? | 7 | 1.1 | 139 | 22.3 | 477 | 76.6 |
| Do you smoke tobacco? | 0 | 0 | 2 | 0.3 | 621 | 99.7 |

**Table 8. Bi-variable and multi-variable logistic regression predicting knowledge related to behavioral risk factors for type 2 diabetes mellitus among women reproductive age group of Gamo zone, Arba Minch towns, 2022 (N = 623).**

| Variables | Categories | BRF knowledge level | | | | COR [CI 95%] | AOR [CI 95%] | P-value |
|---|---|---|---|---|---|---|---|---|
| | | Good | | Poor | | | | |
| | | No | % | No | % | | | |
| Age | 15–24 | 99 | 15.9 | 144 | 23.1 | 1 | 1 | |
| | 25–30 | 78 | 12.5 | 99 | 15.9 | 1.15[0.77–1.69] | 0.58[0.31–1.07] | 0.079 |
| | 31–40 | 88 | 14.1 | 70 | 11.2 | 1.83[1.22–2.74]* | 0.79[0.41–1.54] | 0.495 |
| | > = 41 | 28 | 4.5 | 17 | 2.7 | 2.39[1.25–4.61]* | 0.61[0.24–1.57] | 0.306 |
| Average Family income /birr/ | < 1999 | 52 | 8.3 | 109 | 17.5 | 1 | 1 | |
| | 2000–3000 | 74 | 6.9 | 85 | 7.7 | 1.83[1.16–2.87] | 1.59[0.81–3.14] | 0.174 |
| | 3001–5000 | 84 | 18.5 | 80 | 18.8 | 2.20[1.40–3.45] | 1.81[1.03–3.18]* | 0.039 |
| | > = 5001 | 83 | 13.3 | 56 | 9 | 3.11[1.94–4.99]* | 1.93[1.02–3.68]* | 0.044 |
| Educational status | Illiterate | 11 | 1.8 | 16 | 2.6 | 1 | 1 | |
| | Read and write | 15 | 2.4 | 12 | 1.9 | 1.82[0.62–5.35] | 1.88[0.50–7.12] | 0.355 |
| | Grade1-4 | 15 | 2.4 | 20 | 3.2 | 1.09[0.39–3.02] | 1.02[0.28–3.72] | 0.975 |
| | Grade 5–8 | 42 | 6.7 | 79 | 12.7 | 0.77[0.33–1.82] | 0.87[0.29–2.56] | 0.794 |
| | Grade 9–12 | 72 | 11.6 | 106 | 17 | 0.99[0.43–2.25] | 1.05[0.36–3.06] | 0.933 |
| | Collage & above | 138 | 23.2 | 97 | 15.6 | 2.07[0.92–4.65] | 1.43[0.46–4.39] | 0.573 |
| Occupation | House wife | 72 | 11.6 | 92 | 14.8 | 1 | 1 | |
| | Student | 76 | 12.2 | 107 | 17.2 | 0.91[0.59–1.39] | 0.74[0.37–1.50] | 0.408 |
| | Merchant | 38 | 6.1 | 46 | 7.4 | 1.06[0.62–1.79] | 0.94[0.49–1.81] | 0.845 |
| | Governmental employee | 84 | 13.5 | 44 | 7.1 | 2.44[1.51–3.93]* | 1.74[0.85–3.58] | 0.128 |
| | Private Employee | 14 | 2.2 | 12 | 1.9 | 1.49[0.65–3.42] | 0. 97 [0.33–2.80] | 0.951 |
| | Unemployed | 29 | 4.7 | 9 | 1.7 | 0.40[0.18–0.89] | 0.53[0.19–1.51] | 0.234 |
| Heard about DM | No | 23 | 3.7 | 32 | 5.1 | 1 | 1 | |
| | Yes | 270 | 43.3 | 23 | 3.7 | 18.00[5.58–58.57] | 1.59[0.75–3.34] | 0.224 |
| Source of information | Media | 71 | 12.5 | 114 | 19.9 | 1 | 1 | |
| | Health worker | 71 | 12.5 | 45 | 7.9 | 2.53[1.57–4.08] | 1.87[1.04–3.34] | 0.036 |
| | Friends/relatives | 148 | 26.1 | 119 | 21 | 1.99[1.36–2.92] | 1.65[1.04–2.62] | 0.034* |
| Friend or relative with DM | I don't know | 8 | 1.3 | 44 | 7.10 | 1 | 1 | |
| | No | 137 | 22 | 258 | 41.4 | 1.86[0.83–4.19] | 1.58[0.66–3.79] | 0.303 |
| | Yes | 148 | 23.2 | 28 | 4.50 | 11.77[5.01–27.68] | 4.03[1.56–10.47] | 0.004* |
| Family history of DM | No | 172 | 27.6 | 317 | 50.9 | 1 | 1 | |
| | Yes | 121 | 19.4 | 13 | 2.10 | 17.15[9.40–31.30] | 9.47[4.74–18.90] | 0.000** |

** Significant predictable variable with statistically significant at p<0.001.

* Statistical significant with p Value >0.001

## Discussion

This community-based quantitative cross-sectional study was conducted in Arba Minch town to determine the knowledge of BRF and associated factors regarding T2-DM among WRA.

The study found that 293 (47.0%) [95% CI, 43.5–50.9] participants had good knowledge of the BRF of T2-DM. This finding aligns with previous research conducted in Bale (48%), Nepal (48%), China (48.3%), and Tanzania (48.9%), respectively [25, 28–30]. A possible explanation for this similarity is that many participants might hear it from their friends or relatives. However, this level of knowledge is still lower than that reported in studies conducted in Iran [31], Pakistan [32], Poland [33], Malaysia [18, 34], Nigeria [26], and Debrebrhan [35]. This might be due to the differences in the study populations, which comprise diabetes patients and their family members. Conversely, this result is higher than the study conducted in Stalowa Wola, Poland [30%] [36], Saudi Arabia [29.6%-42.6%] [16, 19], 25.4% in rural Tanzania [21], 19.6% in south-western Nigeria [15], Western Rajasthan, India [17.6%] [37], Bangladesh [13%] [22], 17.3% in Mekele, and 5% in West Go jam [23, 24], Nepal [5.5–20%] [29, 38], and Japan [2.47%] [39]. These inconsistencies can be attributed to the socio-cultural differences among the studied populations. Knowledge of BRF is also a prerequisite for preventing diabetes in the reproductive age group.

Participants with the average family incomes over 3,000 ETH were 1.81 times more likely to possess knowledge than those with incomes below 1,999 ETH. This indicates that those with higher incomes have better access to the necessary knowledge. Overall, the level of knowledge tends to increase with income; as income levels raise, knowledge levels also tend to rise. However, this result contradicts other research from Pakistan [32] and Bale Goba [25]. The difference may be attributed to variations in study setups and socioeconomic status. However, it was not found to be significant in South India [20].

Among the study participants with a family history of DM, the odds of having good knowledge about the BRF of T2-DM were 9.47 times higher compared to those without a family history of DM. This increased awareness may be from a heightened sense of vulnerability and susceptibility to the disease among individuals with a family history. This finding aligns with previous studies conducted in Mekele [30], New York [17], Bangladesh [18], Malaysia [34], Saudi Arabia [29], Nepal [38], and South India [20].

The study also showed an association between having a friend or relative with DM history and knowledge of BRF, a finding consistent with previous research in Jordan and China [28, 40], respectively. Those who had a friend or relative with DM were four times more likely to have BRF knowledge than their counterparts. This may occur when individuals develop a perception of illness's seriousness by discussing it openly with friends or relatives, which can enhance their understanding of risky behaviors.

The study revealed that individuals who received information from health workers had 1.87 times more knowledge than those who obtained it from media. This may be because information from health workers tends to be more comprehensive to understand the BRF. On the other hand, those who acquired information from friends or relatives were 1.65 times more likely to have good knowledge. This could be attributed to the fact that discussing information in an intimate setting with close ones allow for repeated exposure to the key points, reinforcing their understanding. This finding is consistent with a previous study conducted in China [28]. However, a study in Bangladesh did not find a significant association between information from friends or relatives with good diabetes knowledge [22]. This could be due to prevailing negative perception of chronic diseases.

## Strength and limitation of the study

The study has strengths such as a large sample size and a community-based approach. However, its limitations include reliance on literature focused on diabetic patients and difficulties in establishing causal relationships due to its cross-sectional design.

## Conclusion

This community-based cross-sectional study revealed that the overall knowledge about Behavioral Risk Factors (BRF) of Type 2 Diabetes Mellitus (T2-DM) is 47%. A strong statistically significant association was found between average family incomes over 3000 ETH, family history of diabetes, diabetes in a friend or relative, and diabetes information from friends or relatives, as well as health workers with good knowledge of BRF.

A low knowledge score may indicate a potential knowledge gap. Women, particularly those in the reproductive age group, need to have an in-depth understanding of BRF to improve their health by minimizing the risk of T2-DM and its consequences. Women need more access to information on T2-DM risk factors. Additionally, fostering regular community-based health education programs is essential for improving their understanding.

### Recommendations

Focusing on women's knowledge is the best way to reduce the BRF of T2-DM among generations.

### For women reproductive age groups

Women and girls should adopt healthy lifestyles and have equal access to knowledge to protect themselves and their families from T2-DM.

### For health care providers

Regular health education for vulnerable women is essential for ensuring they receive accurate information.

### For programme designers'/policy makers

Women of reproductive age should be encouraged to adopt healthier lifestyles by increasing their awareness of behavioral risks. A nationwide program should also focus on enhancing strategies to mitigate risk factors as a community health priority.

### For the other researchers

In the future, it will be better to assess the risk factors for reproductive-age women using biochemical profiles (blood glucose level, cholesterol level) and anthropometric measurements (blood pressure, body mass index, as well as waist circumference).

Researchers will assess parity among women of reproductive age with knowledge of BRF, as well as their perceptions towards it using the health belief model.

## Acknowledgments

The authors would like to thank Arba Minch College of Health Science for initiating this study. Deep appreciation has gone to Arba Minch general hospital, administration office, Arba Minch town health office, and all administrators of the selected kebeles that were included in

the study. Special thanks to all respondents, data collectors, supervisors, and all other people who were involved in the study directly or indirectly.

## Author Contributions

**Conceptualization:** Tinsae Seyoum, Selamnesh Tesfaye, Yohannes Shiferaw, Rahel Hailu, Dagim Tefera, Zeleke Gebru.

**Data curation:** Tinsae Seyoum, Selamnesh Tesfaye, Yohannes Shiferaw, Rahel Hailu, Dagim Tefera, Zeleke Gebru.

**Formal analysis:** Tinsae Seyoum, Selamnesh Tesfaye, Yohannes Shiferaw, Rahel Hailu, Dagim Tefera, Zeleke Gebru.

**Funding acquisition:** Tinsae Seyoum, Selamnesh Tesfaye, Yohannes Shiferaw, Rahel Hailu, Dagim Tefera.

**Investigation:** Tinsae Seyoum, Selamnesh Tesfaye, Yohannes Shiferaw, Rahel Hailu, Dagim Tefera, Zeleke Gebru.

**Methodology:** Tinsae Seyoum, Selamnesh Tesfaye, Yohannes Shiferaw, Rahel Hailu, Dagim Tefera, Zeleke Gebru.

**Project administration:** Tinsae Seyoum, Selamnesh Tesfaye, Yohannes Shiferaw, Rahel Hailu, Dagim Tefera.

**Resources:** Tinsae Seyoum, Selamnesh Tesfaye, Yohannes Shiferaw, Rahel Hailu, Dagim Tefera.

**Software:** Tinsae Seyoum, Selamnesh Tesfaye, Yohannes Shiferaw, Rahel Hailu, Dagim Tefera, Zeleke Gebru.

**Supervision:** Tinsae Seyoum, Selamnesh Tesfaye, Yohannes Shiferaw, Rahel Hailu, Dagim Tefera, Zeleke Gebru.

**Validation:** Tinsae Seyoum, Selamnesh Tesfaye, Yohannes Shiferaw, Rahel Hailu, Dagim Tefera, Zeleke Gebru.

**Visualization:** Tinsae Seyoum, Selamnesh Tesfaye, Yohannes Shiferaw, Rahel Hailu, Dagim Tefera, Zeleke Gebru.

**Writing – original draft:** Tinsae Seyoum.

**Writing – review & editing:** Tinsae Seyoum, Selamnesh Tesfaye, Yohannes Shiferaw, Rahel Hailu, Dagim Tefera, Zeleke Gebru.

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
