## [Decision Letter · Decision Letter 0]

22 Jan 2024

PONE-D-22-26162Knowledge of behavioural risk factors for types 2 diabetes mellitus and its associated factors among reproductive-age women in Arba Minch town, Gamo zone, 2022PLOS ONE

Dear Dr. seyoum,

Thank you for submitting your manuscript to PLOS ONE. After careful consideration, we feel that it has merit but does not fully meet PLOS ONE’s publication criteria as it currently stands. Therefore, we invite you to submit a revised version of the manuscript that addresses the points raised during the review process.

Please submit your revised manuscript with by Mar 07 2024 11:59PM. If you will need more time than this to complete your revisions, please reply to this message or contact the journal office at plosone@plos.org. Please include the following items when submitting your revised manuscript:A rebuttal letter that responds to each point raised by the academic editor and reviewer(s). You should upload this letter as a separate file labeled 'Response to Reviewers'.A marked-up copy of your manuscript that highlights changes made to the original version. You should upload this as a separate file labeled 'Revised Manuscript with Track Changes'.An unmarked version of your revised paper without tracked changes. You should upload this as a separate file labeled 'Manuscript'.

We look forward to receiving your revised manuscript.

Kind regards,

Liknaw Bewket Bewket Zeleke, Masters

Academic Editor

PLOS ONE

Journal Requirements:

2. You indicated that you had ethical approval for your study. In your Methods section, please ensure you have also stated whether you obtained consent from parents or guardians of the minors (participants below the age of 18 years) included in the study or whether the research ethics committee or IRB specifically waived the need for their consent.

4. Please include your tables as part of your main manuscript and remove the individual files. Please note that supplementary tables (should remain/ be uploaded) as separate "supporting information" files

Additional Editor Comments:

Reviewer 1

This is an important article on knowledge of behavioral risk factors of Diabetes among reproductive age group women in Ethiopia. However, the manuscript needs editing throughout in terms of English language, typos, abbreviations and the result section needs to be articulated in a proper fashion. Thanks

Reviewer 2

I was kindly asked to review the manuscript of Seyoum T et al. I thank the authors for their work, as the result is interesting and achieved through a very well described and reproducible methodological rigor. I believe the article may also be of interest to readers of the journal, but before publication, it should, in my opinion, be fixed in form, following the suggestions given below.

General considerations:

Editing for English language is recommended.

Line numbers in the manuscript file are missing.

Title:

I suggest deleting “2022” at the end of the title and using “…in Arba Minch, Gamo zone, Southern Ethiopia” here and everywhere into the main text, abstract included.

Corresponding author:

Please, report on the same line the corresponding author and the email address

Authorship:

Please, verify with the Editor that, beside a first joint authorship, another equal contribution is admitted.

Please, delete the #aArba Minch, South Nation Nationality Region, Ethiopia, as affiliations report this information.

Abstract, page 2:

Define abbreviations upon first appearance in the text. For example, the abbreviations AOR, BRF and DM are cited, but not defined.

Please, verify the order of words at the end of the Objective sentence and as suggested previously.

Introduction:

Line 2, please add a space between diabetes and (T2DM)

Line 28, please remove the space at the beginning of the sentence

Line 32, please define the abbreviation “RA” before using it

Line 35-38, please evaluate to move this paragraph before the sentence “As a result, …” at line 31 or in the discussions section

Materials and methods:

Please, define the study design: qualitative, quantitative or mix-method?

Please, correct the study period: from February 20, 2022 to March 22, 2022, not February20/2022.

Please, correct “women of reproductive age”. In my personal opinion, a woman of reproductive age is 46 years old, at maximum. I strongly recommend to define the fertile age among inclusion criteria, exclude from the study women of more than 46 years old and revise the manuscript accordingly.

What about parity? It should be of interest to analyse wherever parity >0 may be associated with higher or lower consciousness

Results:

I suggest renaming sub-headings or numbering the entire section as “5.RESULTS” and the others of the manuscript accordingly.

Discussion and Conclusion:

Section are numbered. Either all sections are numbered, or no section is numbered, your choice!

References:

Please, use proper formatting and the same font.

What does it mean “uncategorized references”?

Reviewer 3

Introduction

The manuscript didn’t give a clear idea of the central question that the research is intended to answer and its justification.

•Make sure you provided a sufficient detail on the magnitude of T2DM among reproductive-age women and the consequences for those affected

•The outcome of interest “Knowledge of BRF for T2DM” should have been the central idea in a discussion of why certain factors need more investigation if the problem is to be fully understood. Rather it was "BRF-T2DM" " "behavioral changes -Knowledge" BRF Knowledge - T2DM prevention"…

•The argument on non-existence of studies on this topic in Ethiopia was inconsistent with the fact that numerous studies were used to compare and contrast findings later on the discussion section.

•The statement on significance of the study doesn’t identify the beneficiaries and the benefits of this specific study clearly.

Materials and Methods

The manuscript contain errors in the design and conduct of research.

For example, the statement on

•Sample size determination was incorrect given their statistical implications. “q = proportion of people”

•A multi-stage sampling strategy doesn’t provide sufficient details on the sampling units and sample allocation and selection procedure.

•Variable of the study doesn't identify all the variables used in the analysis including Awareness on DM, Knowledge of Type-2DM, Knowledge of symptoms of Type 2-DM, Knowledge relate to prevention of DM and its complication, Behavioral risk Perception for Type 2DM, Adopting healthy life style for Type 2 DM.

•Data collection was incorrect given their statistical implications “a pre-test of 5 % of the sample population out of the study area” and also described repeatedly. “after sampling technique” and “under a section titled data collection”

•Data processing and analysis was incorrect with regard to description of

•Statistical program used and their purpose. “Epi Data 4.6”

•Statistical procedures used to modify raw data before analysis. “Combining wrong or uncertain responses in determining level of knowledge of the behavioral risk of T2DM”

•Variables used in the analysis “demographic factors” and “the level of knowledge of the behavioral risk of T2DM”

•Summary statistics and graphical techniques used in descriptive statistics for each variables.

•Methods used for analysis coherently. “Binary logistic regression analysis” “multivariable logistic regression” “Descriptive statistic”

•Purpose and reporting format to be followed for the analysis made “Binary logistic regression analysis to identify determinants of behavioural risk factors for T2DM… crude odds ratio (COR) and adjusted odds ratio (AOR) with its respective 95% confidence interval (CI) was used to interpret the result. Then, multivariable logistic…”

Results

The manuscript contain errors in the application, analysis, interpretation, and reporting of statistics. For example

•Age was with different measures that requires difference in properties of distribution. “minimum age of respondents is 15-year-old and maximum age is 49 years” and “mean age of respondents was 27.3 (SD+8.2 years).”

•Continuous variable was also reported without measures of variance. “Average family income was 3910.00 Eth birr(--- SD)”

•Inconsistent format was used in reporting distribution of variables. 235 (37.7%) were collage and above, “married (57.6%)”, “Half of the participants (312, 50.1%) knew T2DM”

•The interpretation of results disregard group sizes for each analysis in many instances in the manuscript. “Source of information (Health workers -116)”, “Have got DM health education (Yes- 159)”

•The text report of the regression analysis states a measure of precision (a confidence interval and P values) for each explanatory variable which is unnecessary.

•Erroneous interpretation of result was made in some instances. “A household with an average income of more than 5,001 Eth. Birr was also 1.93 times...”

•There is no report of whether the variables were assessed for interaction.

Discussion

•The discussion was unjustified, inappropriate, and erroneous in many instances in the manuscript. “The knowledge gap in the WRA may worsen the burden of the illness.”

•The discussion was also made for each explanatory variable which fails to show association with outcome variable unnecessarily.

Conclusions

•Conclusions are presented in inappropriate way. “Nearly 60% of the respondents had no idea about others T2DM risk factors”

•The conclusions was also overstated and erroneously discussed possible implications of the results out of the context of data presented in the manuscript. “This might indicate that women are ignoring their health.”

•The conclusions didn’t provide sufficient details on the association of interest that is “BRF Knowledge” and “Friend /relatives” “Family” “Health worker”

General comment

The language in this manuscript was difficult to understand in many instances and includes.

•Grammatical error: for instance ".. is rises.."

•Typographical error such as "from he study period"

•Unconventional and inappropriate use of abbreviations "RA"

•Issues with substantial clinical implications for example " T2DM is evolving"

In conclusion

•These errors are serious enough to question the conclusions.

•I recommend authors to read PLOS ONE manuscript submission guidelines and seek editorial help in research report write-up.

Reviewers' comments:

Reviewer's Responses to Questions

**Comments to the Author**

1. Is the manuscript technically sound, and do the data support the conclusions?

Reviewer #1: Yes

Reviewer #2: Yes

Reviewer #3: Partly

2. Has the statistical analysis been performed appropriately and rigorously? 

Reviewer #1: Yes

Reviewer #2: Yes

Reviewer #3: No

3. Have the authors made all data underlying the findings in their manuscript fully available?

Reviewer #1: Yes

Reviewer #2: Yes

Reviewer #3: Yes

4. Is the manuscript presented in an intelligible fashion and written in standard English?

Reviewer #1: No

Reviewer #2: Yes

Reviewer #3: No

5. Review Comments to the Author

Reviewer #1: This is an important article on knowledge of behavioral risk factors of Diabetes among reproductive age group women in Ethiopia. However, the manuscript needs editing throughout in terms of English language, typos, abbreviations and the result section needs to be articulated in a proper fashion. Thanks

Reviewer #2: I was kindly asked to review the manuscript of Seyoum T et al. I thank the authors for their work, as the result is interesting and achieved through a very well described and reproducible methodological rigor. I believe the article may also be of interest to readers of the journal, but before publication, it should, in my opinion, be fixed in form, following the suggestions given below.

General considerations:

Editing for English language is recommended.

Line numbers in the manuscript file are missing.

Title:

I suggest deleting “2022” at the end of the title and using “…in Arba Minch, Gamo zone, Southern Ethiopia” here and everywhere into the main text, abstract included.

Corresponding author:

Please, report on the same line the corresponding author and the email address

Authorship:

Please, verify with the Editor that, beside a first joint authorship, another equal contribution is admitted.

Please, delete the #aArba Minch, South Nation Nationality Region, Ethiopia, as affiliations report this information.

Abstract, page 2:

Define abbreviations upon first appearance in the text. For example, the abbreviations AOR, BRF and DM are cited, but not defined.

Please, verify the order of words at the end of the Objective sentence and as suggested previously.

Introduction:

Line 2, please add a space between diabetes and (T2DM)

Line 28, please remove the space at the beginning of the sentence

Line 32, please define the abbreviation “RA” before using it

Line 35-38, please evaluate to move this paragraph before the sentence “As a result, …” at line 31 or in the discussions section

Materials and methods:

Please, define the study design: qualitative, quantitative or mix-method?

Please, correct the study period: from February 20, 2022 to March 22, 2022, not February20/2022.

Please, correct “women of reproductive age”. In my personal opinion, a woman of reproductive age is 46 years old, at maximum. I strongly recommend to define the fertile age among inclusion criteria, exclude from the study women of more than 46 years old and revise the manuscript accordingly.

What about parity? It should be of interest to analyse wherever parity >0 may be associated with higher or lower consciousness

Results:

I suggest renaming sub-headings or numbering the entire section as “5.RESULTS” and the others of the manuscript accordingly.

Discussion and Conclusion:

Section are numbered. Either all sections are numbered, or no section is numbered, your choice!

References:

Please, use proper formatting and the same font.

What does it mean “uncategorized references”?

Reviewer #3: Introduction

The manuscript didn’t give a clear idea of the central question that the research is intended to answer and its justification.

•Make sure you provided a sufficient detail on the magnitude of T2DM among reproductive-age women and the consequences for those affected

•The outcome of interest “Knowledge of BRF for T2DM” should have been the central idea in a discussion of why certain factors need more investigation if the problem is to be fully understood. Rather it was "BRF-T2DM" " "behavioral changes -Knowledge" BRF Knowledge - T2DM prevention"…

•The argument on non-existence of studies on this topic in Ethiopia was inconsistent with the fact that numerous studies were used to compare and contrast findings later on the discussion section.

•The statement on significance of the study doesn’t identify the beneficiaries and the benefits of this specific study clearly.

Materials and Methods

The manuscript contain errors in the design and conduct of research.

For example, the statement on

•Sample size determination was incorrect given their statistical implications. “q = proportion of people”

•A multi-stage sampling strategy doesn’t provide sufficient details on the sampling units and sample allocation and selection procedure.

•Variable of the study doesn't identify all the variables used in the analysis including Awareness on DM, Knowledge of Type-2DM, Knowledge of symptoms of Type 2-DM, Knowledge relate to prevention of DM and its complication, Behavioral risk Perception for Type 2DM, Adopting healthy life style for Type 2 DM.

•Data collection was incorrect given their statistical implications “a pre-test of 5 % of the sample population out of the study area” and also described repeatedly. “after sampling technique” and “under a section titled data collection”

•Data processing and analysis was incorrect with regard to description of

•Statistical program used and their purpose. “Epi Data 4.6”

•Statistical procedures used to modify raw data before analysis. “Combining wrong or uncertain responses in determining level of knowledge of the behavioral risk of T2DM”

•Variables used in the analysis “demographic factors” and “the level of knowledge of the behavioral risk of T2DM”

•Summary statistics and graphical techniques used in descriptive statistics for each variables.

•Methods used for analysis coherently. “Binary logistic regression analysis” “multivariable logistic regression” “Descriptive statistic”

•Purpose and reporting format to be followed for the analysis made “Binary logistic regression analysis to identify determinants of behavioural risk factors for T2DM… crude odds ratio (COR) and adjusted odds ratio (AOR) with its respective 95% confidence interval (CI) was used to interpret the result. Then, multivariable logistic…”

Results

The manuscript contain errors in the application, analysis, interpretation, and reporting of statistics. For example

•Age was with different measures that requires difference in properties of distribution. “minimum age of respondents is 15-year-old and maximum age is 49 years” and “mean age of respondents was 27.3 (SD+8.2 years).”

•Continuous variable was also reported without measures of variance. “Average family income was 3910.00 Eth birr(--- SD)”

•Inconsistent format was used in reporting distribution of variables. 235 (37.7%) were collage and above, “married (57.6%)”, “Half of the participants (312, 50.1%) knew T2DM”

•The interpretation of results disregard group sizes for each analysis in many instances in the manuscript. “Source of information (Health workers -116)”, “Have got DM health education (Yes- 159)”

•The text report of the regression analysis states a measure of precision (a confidence interval and P values) for each explanatory variable which is unnecessary.

•Erroneous interpretation of result was made in some instances. “A household with an average income of more than 5,001 Eth. Birr was also 1.93 times...”

•There is no report of whether the variables were assessed for interaction.

Discussion

•The discussion was unjustified, inappropriate, and erroneous in many instances in the manuscript. “The knowledge gap in the WRA may worsen the burden of the illness.”

•The discussion was also made for each explanatory variable which fails to show association with outcome variable unnecessarily.

Conclusions

•Conclusions are presented in inappropriate way. “Nearly 60% of the respondents had no idea about others T2DM risk factors”

•The conclusions was also overstated and erroneously discussed possible implications of the results out of the context of data presented in the manuscript. “This might indicate that women are ignoring their health.”

•The conclusions didn’t provide sufficient details on the association of interest that is “BRF Knowledge” and “Friend /relatives” “Family” “Health worker”

General comment

The language in this manuscript was difficult to understand in many instances and includes.

•Grammatical error: for instance ".. is rises.."

•Typographical error such as "from he study period"

•Unconventional and inappropriate use of abbreviations "RA"

•Issues with substantial clinical implications for example " T2DM is evolving"

In conclusion

•These errors are serious enough to question the conclusions.

•I recommend authors to read PLOS ONE manuscript submission guidelines and seek editorial help in research report write-up.

6. PLOS authors have the option to publish the peer review history of their article (what does this mean?). If published, this will include your full peer review and any attached files.

Reviewer #1: No

Reviewer #2: No

Reviewer #3: No

---

## [Decision Letter · Decision Letter 1]

6 Sep 2024

PONE-D-22-26162R1Knowledge of behavioral risk factors for type 2 diabetes mellitus and its associated factors among women of reproductive agePLOS ONE

Dear Dr. seyoum,

Thank you for submitting your manuscript to PLOS ONE. After careful consideration, we feel that it has merit but does not fully meet PLOS ONE’s publication criteria as it currently stands. Therefore, we invite you to submit a revised version of the manuscript that addresses the points raised during the review process.

We look forward to receiving your revised manuscript.

Kind regards,

Tomislav Bulum

Academic Editor

PLOS ONE

Reviewers' comments:

Reviewer's Responses to Questions

**Comments to the Author**

1. If the authors have adequately addressed your comments raised in a previous round of review and you feel that this manuscript is now acceptable for publication, you may indicate that here to bypass the “Comments to the Author” section, enter your conflict of interest statement in the “Confidential to Editor” section, and submit your "Accept" recommendation.

Reviewer #1: All comments have been addressed

Reviewer #4: (No Response)

2. Is the manuscript technically sound, and do the data support the conclusions?

Reviewer #1: Yes

Reviewer #4: Partly

3. Has the statistical analysis been performed appropriately and rigorously? 

Reviewer #1: Yes

Reviewer #4: No

4. Have the authors made all data underlying the findings in their manuscript fully available?

Reviewer #1: Yes

Reviewer #4: Yes

5. Is the manuscript presented in an intelligible fashion and written in standard English?

Reviewer #1: Yes

Reviewer #4: No

6. Review Comments to the Author

**Reviewer #1: **(No Response)

**Reviewer #4: **Thank you for the opportunity to review this interesting manuscript on women's knowledge about DM type 2 risk factors.

The information is important from the public health point of view, both regional and perhaps world-wide. However, There are some points to be addressed:

1. An English proofing tool should be used, as there are some sentences throughout the text which are not properly constructed.

2. Authors should provide some information regarding the representativity of the sample for the Ethiopian female population.

3. The results are a tad difficult to follow and the list of variables is quite long. The logistic regression model is not quite clearly presented.

4. The Discussion section should be improved - the comparison with other studies should be a tad more detailed, especially with the Ethiopian ones. Also, the results are partly repeated, but not discussed.

5. The conclusions are not quite conclusions - there is a mixture of discussions and general statements.

7. PLOS authors have the option to publish the peer review history of their article (what does this mean?). If published, this will include your full peer review and any attached files.

Reviewer #1: No

Reviewer #4: No

---

## [Author Response · Author response to Decision Letter 1]

10 Nov 2024

RESPONSE TO REVIEWER

[Cover Letter] 

Dear Editor,

We genuinely appreciate you and the reviewers for taking the time to review our paper and provide valuable comments. Your insightful feedback has significantly contributed to the improvements in the current version. We have carefully considered each comment and have made every effort to address them. We hope that the revised manuscript meets your high standards. We also welcome any further constructive comments. Below, we provide point-by-point responses. All modifications in the manuscript have been highlighted in blue on a white background.

Sincerely,

Tinsae Seyoum (PI)

tswy2006@gmail.com

+251913260781

Lecturer, Department of Public Health

Arba Minch College of Health Sciences

[General comments] ‘’When submitting your revision, we need you to address these additional requirements.

https://journals.plos.org/plosone/s/file?id=wjVg/PLOSOne_formatting_sample_main_body. pdf and https://journals.plos.org/plosone/s/file?id=ba62/PLOSOne_formatting_sample_title_authors_affiliations.pdf

2. You indicated that you had ethical approval for your study. In your Methods section, please ensure you have also stated whether you obtained consent from parents or guardians of the minors (participants below the age of 18 years) included in the study or whether the research ethics committee or IRB specifically waived the need for their consent.

3. When completing the data availability statement of the submission form, you indicated that you would make your data available on acceptance. We strongly recommend all authors decide on a data sharing plan before acceptance, as the process can be lengthy and hold up publication timelines. Please note that, though access restrictions are acceptable now, your entire data will need to be made freely accessible if your manuscript is accepted for publication. This policy applies to all data except where public deposition would breach compliance with the protocol approved by your research ethics board. If you are unable to adhere to our open data policy, please kindly revise your statement to explain your reasoning, and we will seek the editor's input on an exemption. Please be assured that, once you have provided your new statement, the assessment of your exemption will not hold up the peer review process.

4. Please include your tables as part of your main manuscript and remove the individual files. Please note that supplementary tables should remain or be uploaded as separate "supporting information" files

Response:

[General Comment] ‘’This is an important article on knowledge of behavioral risk factors of Diabetes among reproductive age group women in Ethiopia. However, the manuscript needs editing throughout in terms of English language, typos, and abbreviations, and the result section needs to be articulated in a proper fashion. Thanks’’

Response to Reviewer 1

Response: Thanks for your valuable comments. We revised the English language and typos and abbreviations that needed clarification. We tried to articulate the result section in the proper way as much as possible for the entire paper (from the abstract to the conclusion page).

Response to Reviewer 2

[General Comment] ‘’I was kindly asked to review the manuscript of Seyoum T et al. I thank the authors for their work, as the result is interesting and achieved through a very well described and reproducible methodological rigor. I believe the article may also be of interest to readers of the journal, but before publication, it should, in my opinion, be fixed in form, following the suggestions given below.’’

Response: Thank you very much for your comments that helped us improve this manuscript

Comment 1] ‘’Editing for English language is recommended. Line numbers in the manuscript file are missing.’’

Response: Thanks for your kind reminders. We edited almost all sentences thoroughly and provided line numbers in the whole manuscript file.

[Comment 2] Title:

‘’I suggest deleting “2022” at the end of the title and using "in Arba Minch, Gamo zone, Southern Ethiopia” here and everywhere into the main text, abstract included.’’

Response: Thanks. We edited the title page, abstract and the main texts by removing ‘’2022’’ and’’ in Arba Minch, Gamo zone, South Ethiopia’’ except from the tables and figures. 

The revised Title: Knowledge of behavioral risk factors for type 2 diabetes mellitus and its associated factors among reproductive-age women on page 1, lines 1–3 

Abstract: Objectives, To assess knowledge of the behavioral risks of type 2 diabetes mellitus and its associated factors among women of reproductive age./page 2, line 22-23

[Comment 3] ‘’Corresponding author:

Please report on the same line the corresponding author and the email address.’’

Response: Thanks for your kind reminders. We edited accordingly.

*= Corresponding author: Email address, tswy2006@gmail.com, page no. 1, line 6

[Comment 4] ‘’Authorship:

Please verify with the Editor that, beside a first joint authorship, another equal contribution is admitted.

Please delete the #aArba Minch, South Nation Nationality Region, Ethiopia, as affiliations report this information.’’

Response: Thanks a lot for your comment. We verified the equal contribution of the authors.

We removed this symbol'#@' and added these symbols¶, & on equal contribution based on PLOSE rule.

We removed the sentence '#a, Arba Minch, South Nation Nationality Region, Ethiopia.' from the cover page.(Page 1) 

[Comment 5] ‘’Abstract, page 2:

Define abbreviations upon first appearance in the text. For example, the abbreviations AOR,

BRF and DM are cited but not defined.

Please verify the order of words at the end of the objective sentence and as suggested previously.’’

Response: Thank you for your nice suggestion. We revised the abbreviation on the abstract page2, lines no. 30, 31, 32, and 39: Adjusted Odd Ratio, Behavioral Risk Factors, and Diabetes Mellitus.”

The objective was also amended according to your suggestions and comments by removing ‘2022’ and’ in Arba Minch town, South Ethiopia.’

[Comment 6] ‘’Introduction:

Line 2, please add a space between diabetes and (T2DM)

Line 28, please remove the space at the beginning of the sentence.

Line 32, please define the abbreviation “RA” before using it

Line 35-38, please evaluate to move this paragraph before the sentence “As a result,...” at line

31 or in the discussions section.’’

Response: Thank you very much for the reminder. We have made revisions accordingly.

We detached the paragraph of ‘Knowledge of BRF is also a prerequisite for preventing diabetes in reproductive age groups into the discussion session.

[Comment 7] ‘’Results:

I suggest renaming sub-headings or numbering the entire section as “5.RESULTS” and the ‘’

Response: Thank you for the nice reminder. We renamed the subheading section and removed numbering styles.

[Comment 8] ‘’Materials and methods:

Please define the study design: qualitative, quantitative, or mix-method?

Please correct the study period: from February 20, 2022, to March 22, 2022, not February 20, 2022.

Please correct “women of reproductive age.”. In my personal opinion, a woman of reproductive age is 46 years old, at maximum. I strongly recommend defining the fertile age among inclusion criteria, excluding from the study women of more than 46 years old, and revising the manuscript accordingly.

What about parity? It should be of interest to analyses wherever parity > 0 may be associated with higher or lower consciousness.’’

Response: Thank you for your eminent comments, and we revised accordingly as follows:

 We defined the study design by adding ''quantitative'' on page 4, line 91.

We corrected the study period from February 20, 2022 to March 22, 2022. (Page 4, line no. 92)

We changed the text to “women of reproductive age” in all our text, page1, line 2-3; page 2, lines no 20, 23, 31and line 43 including key words.

Concerning participants’ age, we relied on the data of the WHO and Ethiopian Demography Health Survey (EDHS) to consider that a woman of reproductive age is up to 49 years older than 46, and all analyses were made by age between 15 and 49. Otherwise, if we consider the age up to 46, it could affect the whole data and its result.

Unfortunately, in this study, we didn't consider the ''parity’’ variable, so we recommended for future to be addressed in the recommendation part. (Page 19, line no.. 388-389).

Comment 9] ‘’Discussion and Conclusion:

Sectionss are numbered. Either all sections are numberedor no section is numbered, your choice!’’

Response: Thank you for your nice reminder. We revised by removing numbers from all Sections

[Comment 10] ‘’References:

Pleaseuse proper formatting and the same font.

What does it mean “uncategorized references”?’

Response: Thanks for your suggestion.

We revised the fonts in the reference to be the same at all levels and adjusted the formats by using endnote accordingly, and the uncategorized part was totally removed. 

Response to Reviewer 3

[Comment 1]” Introduction

The manuscript didn’t give a clear idea of the central question that the research is intended to answer and its justification.

Make sure you providedufficient detail on the magnitude of T2DM among reproductive-age women and the consequences for those affected

The outcome of interest “Knowledge of BRF for T2DM” should have been the central idea in a discussion of why certain factors need more investigation if the problem is to be fully understood. Rather it was "BRF-T2DM" " "behavioral changes-knowledge BRF Knowledge - T2DM prevention"…

The argument on the non-existenceence of studies on this topic in Ethiopia was inconsistent with the fact that numerous studies were used to compare and contrast findings later on in the discussion section.

The statement on the significance of the study doesn't identify the beneficiaries and the benefits of this specific study clearly.’’

Response: Thank you very much for pointing this out. We revised the sentence to incorporate the magnitude of T2DM among WRA and its consequences and what this research intendss to answer as follows: [In page 4, lines-82], we rearrange the sentence and add, ‘’T2-DM is experiencing rapid global expansion due to several factors, including aging populations, economic development, and urbanization.''.''. These changes are all associated with behavioral risk factors (BRFs). The incidence of diabetes also increases gradually during puberty, and it is more prevalent in women, possibly due to hormonal variations and insulin resistance (4). Additionally, it is linked to reproductive issues in individuals of reproductive age, which can be addressed through behavioral changes (7). Around 451 million people over the age of 18 have diabetes, and nearly 22 million women’s live births are affected by the prenatal period (8). 

Italso affects fourteen percent of all live births in the Middle East and Africa (9). Preventing diabetes at the population level relies heavily on public awareness and a proper understanding of its risk factors (10).

There are multiple factors contributing to the development of T2-DM, categorized as non-modifiable and modifiable risk factors. The modifiable risk factors known as BRFs that can increase the likelihood of developing T2-DM include obesity, physical inactivity, unhealthy eating habits, smoking, excessive alcohol consumption, insomnia, and stress (11, 12). These issues often remain hidden in countries with low socioeconomic status, including our country (13, 14).

BRF significantly predisposes individuals to T2-DM. However, adopting positive lifestyle changes and reducing negative behaviors can reflect an individual's knowledge of these risks (15). Although different studies have assessed knowledge of certain risk factors like sedentary lifestyle, obesity, poor diet, and lack of exercise in diabetes and chronic disease patients, they have overlooked other factors. This study incorporates all possible factors and addresses the limited understanding of knowledge on BRF (10, 15-22). In Ethiopia, research has predominantly concentrated on the general understanding of DM, with limited evidence on the knowledge regarding BRFs (23-25).’’

We also identified the beneficiaries and the benefits of this specific study on page 4, lines 839

‘’The benefit of this study is improving health outcomes by identifying women's awareness and increasing their knowledge of BRF. It aims to target interventions and inform policy development. The beneficiaries include WRA, health care providers, policymakers families, and communities. 

Therefore, this study assesses WRA's understanding of all potential BRFs for T2-DM and associated factors. This study can also serve as a base for interventions, improve knowledge on healthy lifestyle modifications and T2-DM prevention, and be a reference for scholars.’’

[Comment 2] ‘’Materials and Methods

The manuscript contains errors in the design and conduct of research.

For example, the statement on sample size determination was incorrect given their statistical implications. "q = proportion of people’’

Response: Thanks for your comment. We removed the ‘’ q= proportion of people’’, 

Some text modification was made to be more coherent on page 5, line 100-103, 107, 11313-125

[Comment 3] ‘’A multi-stage sampling strategy doesn’t provide sufficient details on the sampling units and sample allocation and selection procedure.’’

Response: Thanks a lot for these comments. We revised the anuscript and elaboratedd with detail on sampling units and ample selection procedures on page no.. 5, lines 118-123,, as follows: ‘’The computed sample size was proportionally allocated using a multistage sampling method based on the population size in the selected kebelesafter obtaining the number of households from the census. Then,, every 13th household interval, the house was selected until the required sample size was fulfilled. The first house was chosen using the lottery method. If multiple people in a household met the inclusion criteria, only one member was chosen by lottery.’’

[Comment 4] ‘’Variable of the study doesn't identify all the variables used in the analysis, including awareness of DM, knowledge of Type-2DM, and knowledge of symptoms of Type-2DM. Knowledge related to prevention of DM and its complications, behavioral risk perception for Type 2DM, and adopting a healthy lifestyle for Type 2DM.’’

Response: Thanks for your reminder. We added the variable that was identified in the analysis. Page 5, lines no. 124-128. 

We considered knowledge of T2-DM (Symptoms and knowledge related to prevention and complication as general knowledge) and we also included hearing about DM, behavioral risk perception and adopting a healthy life in our analysis, but all of them were insignificant 

[Comment 5] ‘’Data collection was incorrect given their statistical implications, “a pre-test of 5% of the sample population out of the study area,” and also described repeatedly. “after sampling technique” and “under a section titled data collection.”

Data processing and analysis were incorrect with regard to description of Statistical program used and their purpose. “Epi Data 4.6": Statistical procedures used to modify raw data before analysis. “Combining wrong or uncertain responses in determining level of knowledge of the behavioral risk of T2DM”

Variables used in the analysis were “demographic factors” and “the level of knowledge of the behavioral risk of T2DM.”

Summary statistics and graphical techniques used in descriptive statistics for each variable.

Methods used for analysis coherently. “Binary logistic regression analysis,” “multivariable logistic regression,” “descriptive statistic.”

Purpose and reporting format to be followed for the analysis made “Binary logistic regression analysis to identify determinants of behavioural risk factors for T2DM... crude odds ratio (COR) and adjusted odds ratio (AOR) with their respective 95% confidence intervals (CI) were used to determine the number of measurement stations included. Interpret the result. Then, multivariable logistic…” 

---

## [Decision Letter · Decision Letter 2]

2 Dec 2024

PONE-D-22-26162R2Knowledge of behavioral risk factors for type 2 diabetes mellitus and its associated factors among women of reproductive agePLOS ONE

Dear Dr. seyoum,

Thank you for submitting your manuscript to PLOS ONE. After careful consideration, we feel that it has merit but does not fully meet PLOS ONE’s publication criteria as it currently stands. Therefore, we invite you to submit a revised version of the manuscript that addresses the points raised during the review process.

We look forward to receiving your revised manuscript.

Kind regards,

Tomislav Bulum

Academic Editor

PLOS ONE

Reviewers' comments:

Reviewer's Responses to Questions

**Comments to the Author**

1. If the authors have adequately addressed your comments raised in a previous round of review and you feel that this manuscript is now acceptable for publication, you may indicate that here to bypass the “Comments to the Author” section, enter your conflict of interest statement in the “Confidential to Editor” section, and submit your "Accept" recommendation.

Reviewer #4: (No Response)

2. Is the manuscript technically sound, and do the data support the conclusions?

Reviewer #4: Partly

3. Has the statistical analysis been performed appropriately and rigorously? 

Reviewer #4: I Don't Know

4. Have the authors made all data underlying the findings in their manuscript fully available?

Reviewer #4: Yes

5. Is the manuscript presented in an intelligible fashion and written in standard English?

Reviewer #4: No

6. Review Comments to the Author

Reviewer #4: Thank you for submitting a revised version of the manuscript. The information is quite important. However, there are some relevant issues still pending:

- English proofing is still much needed; some of the phrases are poorly constructed; active and passive voice alternates, etc.

- the methods section should be improved - the variables should be clearly defined; the choice for the cut-off for "good knowledge" should be mentioned;

- the discussion part is lacking consistency; it merely mentions other studies/countries with similar studies; should focus more on previous knowledge in Ethiopia, mention the other studies, perhaps comparing the methods used and then the results.

- the conclusion part is rephrased, but restates the results.

I would suggest a major revision of the whole manuscript with the help of an English proofing system and other manuscripts on the subject.

7. PLOS authors have the option to publish the peer review history of their article (what does this mean?). If published, this will include your full peer review and any attached files.

Reviewer #4: No

---

## [Author Response · Author response to Decision Letter 2]

5 Jan 2025

The previous reviewer's general comments 

 [General comments] ‘’When submitting your revision, we need you to address these additional requirements.

https://journals.plos.org/plosone/s/file?id=wjVg/PLOSOne_formatting_sample_main_body.pdf and https://journals.plos.org/plosone/s/file?id=ba62

2. You indicated that you had ethical approval for your study. In your Methods section, please ensure you have also stated whether you obtained consent from parents or guardians of the minors (participants below the age of 18 years) included in the study or whether the research ethics committee or IRB specifically waived the need for their consent.

3. When completing the data availability statement of the submission form, you indicated that you will make your data available on acceptance. We strongly recommend all authors decide on a data sharing plan before acceptance, as the process can be lengthy and hold up publication timelines. Please note that, though access restrictions are acceptable now, your entire data will need to be made freely accessible if your manuscript is accepted for publication. This policy applies to all data except where public deposition would breach compliance with the protocol approved by your research ethics board. If you are unable to adhere to our open data policy, please kindly revise your statement to explain your reasoning, and we will seek the editor's input on an exemption. Please be assured that, once you have provided your new statement, the assessment of your exemption will not hold up the peer review process.

4. Please include your tables as part of your main manuscript and remove the individual files. Please note that supplementary tables (should remain/be uploaded) as separate "supporting information" files.

Response to previous reviewer comments:

Response to general comments 

Response to General Comment 1: Thank you for your comment. We have revised the article title, the symbol legend, and the corresponding author email initials in parentheses based on the PLOS ONE style templates.

Response to General comment 2: Thank you for your valuable comment. We have indicated that the ethical approval for our study obtained consent from parents and guardians of the minors (participants below the age of 18 years) included in the study on page 7, lines 183-187.

Response to General comment 3: Thank you for your comment. We have declared that our entire data will be made freely accessible if this manuscript is accepted for publication.

Response to General Comment 4: Thank you for your nice comment. We have incorporated the tables into the main manuscript and removed the individual file as per the comments.

Response to Reviewer 1

[General Comment] ‘’This is an important article on knowledge of behavioral risk factors of diabetes among reproductive-age group women in Ethiopia. However, the manuscript needs editing throughout in terms of English language, typos, and abbreviations, and the results section needs to be articulated in a proper fashion. Thanks.

Response 1: Thank you for your valuable comments. We have revised the English language, corrected typos, and clarified abbreviations. We have also articulated the results section properly throughout the entire paper, from the abstract to the conclusion.

Response to Reviewer 2

[General Comment 2]’ I was kindly asked to review the manuscript of Seyoum T et al. I thank the authors for their work, as the result is interesting and achieved through a very well described and reproducible methodological rigor. I believe the article may also be of interest to readers of the journal, but before publication, it should, in my opinion, be fixed in form, following the suggestions given below.

Response.2: Thank you very much for your comments that helped us improve this manuscript.

[Comment 2.1] ‘’Editing for English language is recommended. Line numbers in the manuscript file are missing.

Response 2.1: Thanks for your kind reminders. We have edited almost all sentences thoroughly and provided line numbers in the whole manuscript file.

[Comment 2.2] Title:

’I suggest deleting “2022” at the end of the title and using “...in Arba Minch, Gamo zone, Southern Ethiopia” here and everywhere in the main text, abstract included.’’

Response 2.2: Thanks. We edited the title page, abstract, and the main texts by removing’2022" and in Arba Minch, Gamo zone, South Ethiopia, except from the tables and figures. 

The revised title: Knowledge of behavioral risk factors for type 2 diabetes mellitus and its associated factors among reproductive-age women on page 1, lines 1-3 

Abstract: Objectives, To assess knowledge of the behavioral risks of type 2 diabetes mellitus and its associated factors among women of reproductive age. /page 2, line 22-23/

[Comment 2.3] ‘’Corresponding author:

Please report on the same line the corresponding author and the email address.

Response 2.3: Thanks for your kind reminders. Thank you for your kind reminders. We have made the necessary edits accordingly.

*= Corresponding author: Email address, tswy2006@gmail.com (T.S.) on page no. 1, line 12

[Comment 2.4] ‘’Authorship:

Please verify with the editor that, besides a first joint authorship, another equal contribution is admitted.

Please delete the #aArba Minch, South Nation Nationality Region, Ethiopia, as affiliations report this information.

Response 2.4: Thank you for your valuable comment. We have verified the equal contribution of the authors. We removed the symbol '#@' and replaced it with '¶' and '&' for equal contribution, based on the PLOS ONE guidelines. Additionally, we removed the sentence '#a, Arba Minch, South Nation Nationality Region, Ethiopia.' from the cover page (Page 1).

[Comment 2.5] ‘’Abstract, page 2:

Define abbreviations upon first appearance in the text. For example, the abbreviations AOR,

 BRF and DM are cited but not defined.

Please verify the order of words at the end of the objective sentence as suggested previously.

Response 2.5: Thank you for your nice suggestion. We revised the abbreviations on the abstract page 2, lines 30, 31, 32, 34, and 39 to “Adjusted Odd Ratio, Behavioral Risk Factors, and Diabetes Mellitus.” The objective was also amended according to your suggestions and comments by removing ‘2022’ and ‘in Arba Minch town, South Ethiopia.’

[Comment 2.6] ‘’Introduction:

Line 2, please add a space between diabetes and (T2DM).

Line 28, please remove the space at the beginning of the sentence.

Line 32, please define the abbreviation “RA” before using it.

Lines 35-38: Please evaluate moving this paragraph before the sentence “As a result, …” at line 31 or in the discussions section.

Response 2.6: Thank you for your valuable comment. We have made the necessary revisions accordingly. The paragraph stating, 'Knowledge of BRF is also a prerequisite for preventing diabetes in reproductive age groups,' has been moved to the discussion section.

[Comment 2.7]’Results:

I suggest renaming sub-headings or numbering the entire section as “5. RESULTS” and the’

Response 2.7: Thank you for your valuable comment. We have renamed the subheading section and removed the numbering styles as per your suggestion.

[Comment 2.8] ‘’Materials and methods:

Please define the study design: qualitative, quantitative, or mixed-method?

Please correct the study period: from February 20, 2022, to March 22, 2022, not February 20, 2022.

Please correct “women of reproductive age.” In my personal opinion, a woman of reproductive age is 46 years old, at maximum. I strongly recommend defining the fertile age among inclusion criteria, excluding from the study women of more than 46 years old, and revising the manuscript accordingly.

What about parity? It should be of interest to analyses wherever parity > 0 may be associated with higher or lower consciousness.

Response 2.8: Thank you for your eminent comments, and we revised accordingly as follows: Thank you for your eminent comments. We have made the following revisions accordingly:

• We defined the study design by adding "quantitative" on page 4, line 90.

• We corrected the study period to February 20, 2022, to March 22, 2022 (Page 4, line 91).

• We changed the text to "women of reproductive age" throughout the manuscript, including page 1, lines 2-3; WRA on page 2, lines 23, 31, and 43; and in the keywords.

• Regarding the participants' age, we used data from the World Health Organization (WHO) and the Ethiopian Demographic Health Survey (EDHS) to define a woman of reproductive age as being between 15 and 49 years old. All analyses were conducted for ages 15 to 49. Considering age up to 46 would affect the entire data and results.

• Unfortunately, we did not consider the "parity" variable in this study. We have recommended addressing this in future research in the recommendation section (Page 19, lines 389-390).

[Comment 2. 9] ‘’Discussion and Conclusion:

Section are numbered. Either all sections are numbered, or no section is numbered, your choice!’’

[Comment 2. 9] ‘’Discussion and Conclusion:

Section are numbered. Either all sections are numbered, or no section is numbered, your choice!’’

Response 2.9: Thank you for your valuable comment. We have revised the manuscript by removing numbers from all sections as per your suggestion.

[Comment 2.10] ‘’References:

Please, use proper formatting and the same font.

What does it mean “uncategorized references”?’’

Response 2.10: Thank you for your valuable suggestion. We have revised the fonts in the references to ensure consistency at all levels and adjusted the formats using EndNote accordingly. Additionally, the uncategorized part has been completely removed.

Response to Reviewer 3

[Comment 3.1]” Introduction

The manuscript didn’t give a clear idea of the central question that the research is intended to answer and its justification.

Make sure you provided a sufficient detail on the magnitude of T2DM among reproductive-age women and the consequences for those affected

The outcome of interest “Knowledge of BRF for T2DM” should have been the central idea in a discussion of why certain factors need more investigation if the problem is to be fully understood. Rather it was "BRF-T2DM" " "behavioral changes -Knowledge" BRF Knowledge - T2DM prevention"…

The argument on non-existence of studies on this topic in Ethiopia was inconsistent with the fact that numerous studies were used to compare and contrast findings later on the discussion section.

The statement on significance of the study doesn't identify the beneficiaries and the benefits of this specific study clearly.’’

Response 3.1: Thank you very much for pointing this out. We revised the sentence to incorporate the magnitude of T2DM among WRA and its consequences and what this research intend to answer as follows: [In page 3-4, lines no 58-85], we rearrange the sentence and add, ‘’T2-DM is experiencing rapid global expansion due to several factors, including aging populations, economic development, and urbanization. These changes are all associated with BRFs. The incidence of diabetes also increases gradually during puberty, and it is more prevalent in women, possibly due to hormonal variations and insulin resistance (4). Additionally, it is linked to reproductive issues in individuals of reproductive age, which can be addressed through behavioral changes (7). Around 451 million people over the age of 18 have diabetes, and nearly 22 million women’s live births are affected by the prenatal period (8). 

It also affects fourteen percent of all live births in the Middle East and Africa (9). Preventing diabetes at the population level relies heavily on public awareness and a proper understanding of its risk factors (10).

There are multiple factors contributing to the development of T2-DM, categorized as non-modifiable and modifiable risk factors. The modifiable risk factors known as BRFs that can increase the likelihood of developing T2-DM include obesity, physical inactivity, unhealthy eating habits, smoking, excessive alcohol consumption, insomnia, and stress (11, 12). These issues often remain hidden in countries with low socioeconomic status, including our country (13, 14).

BRF significantly predisposes individuals to T2-DM. However, adopting positive lifestyle changes and reducing negative behaviors can reflect an individual's knowledge of these risks (15). Although different studies have assessed knowledge of certain risk factors like sedentary lifestyle, obesity, poor diet, and lack of exercise in diabetes and chronic disease patients, they have overlooked other factors. This study incorporates all possible factors and addresses the limited understanding of knowledge on BRF (10, 15-22). In Ethiopia, research has predominantly concentrated on the general understanding of DM, with limited evidence on the knowledge regarding BRFs (23-25).’’

We also identified the beneficiaries and the benefits of this specific study on page4, line no.83-89

‘’The benefit of this study is improving health outcomes by identifying women's awareness and increasing their knowledge of BRF. It aims to target interventions and inform policy development. The beneficiaries include WRA, health care providers, policy makers, families, and communities. 

Therefore, this study assesses WRA's understanding of all potential BRFs for T2-DM and associated factors. This study can also serve as a base for interventions, improve knowledge on healthy lifestyle modifications and T2-DM prevention, and be a reference for scholars.’’

[Comment 3.2] ‘’Materials and Methods

The manuscript contains errors in the design and conduct of research.

For example, the statement on Sample size determination was incorrect given their statistical implications. “q =proportion of people’’

Response 3.2: Thanks for your comment. We removed the ‘’ q= proportion of people’’, 

Some text modification was made to be more coherent on page 5, line 100-103,104,111-113,116-128

[Comment 3.3] ‘’A multi-stage sampling strategy doesn’t provide sufficient details on the sampling units and sample allocation and selection procedure.’’

Response 3.3: Thanks a lot for these comments. We revised the manuscript and elaborate with detail on sampling units and ample selection procedures on page no 5, lines 118-123 as follows: ‘The sample size was determined using a multistage sampling method, which allocated participants proportionally based on the population size of the selected kebeles. After obtaining the number of households from the census, every 13th household was selected until the required sample size was achieved. The first house was selected using a lottery method. If multiple members of a household met the inclusion criteria, only one individual was chosen through the lottery.’’

[Comment 3.4] ‘’Variable of the study doesn't identify all the variables used in the analysis including Awareness on DM, Knowledge of Type-2DM, Knowledge of symptoms of Type 2-DM, Knowledge relate to prevention of DM and its complication, Behavioral risk Perception for Type 2DM, Adopting healthy life style for Type 2 DM.’’

Response 3.4: Thanks for your reminder. We added the variable that was identified in the analysis. Page 5, lines no. 124-128. 

In this study, knowledge of BRF is the dependent variable. The independent variables include general knowledge of T2-DM (such as its meaning, symptoms, prevention, and control), behavioral risk perception, leading a healthy lifestyle, socio-demographic factors, and other relevant factors encompass personal history of diabetes, family history of diabetes, exposure to diabetes health education, awareness of diabetes, and sources of information.

[Comment 3.5] ‘’Data collection was incorrect given their statistical implications “a pre-test of 5 % of the sample population out of the study area” and also described repeatedly. “after sampling technique” and “under a section titled data collection”

Data processing and analysis was incorrect with regard to description of Statistical program used and their purpose. “Epi Data 4.6” Statistical procedures used to modify raw data before analysis. “Combi

---

## [Editor Report · Decision Letter 3]

14 Jan 2025

Knowledge of behavioral risk factors for type 2 diabetes mellitus and its associated factors among women of reproductive age

PONE-D-22-26162R3

Dear Dr. Seyoum

We’re pleased to inform you that your manuscript has been judged scientifically suitable for publication and will be formally accepted for publication once it meets all outstanding technical requirements.

Kind regards,

Tomislav Bulum

Academic Editor

PLOS ONE

Additional Editor Comments (optional):

Thank you for adressing all comments.

---

## [Editor Report · Acceptance letter]

20 Jan 2025

PONE-D-22-26162R3 

PLOS ONE

Dear Dr. seyoum, 

I'm pleased to inform you that your manuscript has been deemed suitable for publication in PLOS ONE. Congratulations! Your manuscript is now being handed over to our production team.

Kind regards, 

on behalf of

Dr. Tomislav Bulum 

Academic Editor

PLOS ONE